

# Opinion: The strength of long-term comprehensive observations to meet multiple grand challenges at different environments and in the atmosphere

Markku Kulmala[1,2,3], Anna Lintunen[1,4], Hanna Lappalainen[1], Annele Virtanen[5], Chao Yan[1,2,3], Ekaterina Ezhova[1], Tuomo Nieminen[1,6], Ilona Riipinen[7], Risto Makkonen[1,8], Johanna Tamminen[9], Anu-Maija Sundström[9], Antti Arola[8], Armin Hansel[10], Kari Lehtinen[5], Timo Vesala[1], Tuukka Petäjä[1], Jaana Bäck[4], Tom Kokkonen[1] and Veli-Matti Kerminen[1]

[1]Institute for Atmospheric and Earth System Research / Physics, Faculty of Science, University of Helsinki, Finland
[2]Joint International Research Laboratory of Atmospheric and Earth System Sciences, School of Atmospheric Sciences, Nanjing University, Nanjing, China
[3]Aerosol and Haze Laboratory, Beijing Advanced Innovation Center for Soft Matter Science and Engineering, Beijing University of Chemical Technology, Beijing, China
[4]Institute for Atmospheric and Earth System Research / Forest Sciences, Faculty of Agriculture and Forestry, University of Helsinki, Finland
[5]Department of Technical Physics, University of Eastern Finland, Kuopio, Finland
[6]Department of Physics, University of Helsinki, Finland
[7]Bolin Centre for Climate Research, Department of Environmental Science (ACES), Stockholm University, Sweden
[8]Climate Research Programme, Finnish Meteorological Institute, Helsinki, Finland
[9]Space and Earth Observation Centre, Finnish Meteorological Institute, Helsinki, Finland
[10]Institute for Ion and Applied Physics, University of Innsbruck, Austria

*Correspondence to*: Markku Kulmala (markku.kulmala@helsinki.fi)

**Abstract.** To be able to meet global grand challenges (climate change; biodiversity loss; environmental pollution; scarcity of water, food and energy supplies; acidification; deforestation; chemicalization; pandemics), which all are closely interlinked with each other, we need comprehensive open data with proper metadata. The large data sets from ground-base in situ observations, ground and satellite remote sensing and multiscale modelling need to be utilized seamlessly. In this opinion paper, we describe the SMEAR (Station for Measuring Earth surface – Atmosphere Relations) concept. We also demonstrate its power via several examples, such as detection of new particle formation and their subsequent growth, quantifying atmosphere-ecosystem feedback loops, combining comprehensive observations with emergency science and services, as well as studying the effect of COVID restrictions on different air quality and climate variables. The future needs and the potential of comprehensive observations of the environment are summarized.

## 1 Background

The Earth is facing several environmental challenges on a global scale, often called "Grand Challenges" (https://www.wcrp-climate.org/grand-challenges/). The growing population (https://pdp.unfpa.org) needs fresh air, fresh water, food and energy,



while at the same time climate is changing, many cities have challenges with air quality, biodiversity is decreasing and supplies
of fresh water, food and energy are diminishing. Since these Grand Challenges are highly connected and interlinked, not only
with each other but also with e.g. pandemics, they cannot be solved separately since potential solutions are tightly coupled
with each other (e.g. Kulmala et al., 2015; Lappalainen et al., 2016; Nolan et al., 2018; Laughner et al., 2021; Wang et al.,
2023). However, the solutions may also include unexpected trade-offs (e.g. Fastre et al 2020). Therefore, integrated,
comprehensive, big open data are required (Kulmala et al., 2021), together with a research and innovation framework in which
a multidisciplinary research with critical mass of scientists utilising proper resources is connected to fast-tracked policy making
and wide stakeholder community. This allows aiming for practical solutions based on deep scientific understanding.

The global challenges are intimately linked to interactions and feedbacks between the different compartments of the planet
Earth at different spatial and temporal scales. Fundamentally, the atmosphere is closely interconnected with various other parts
of the Earth, including biosphere, hydrosphere, cryosphere and lithosphere as well as urban surfaces over a range of time and
spatial scales varying from seconds to millennia (Wanner et al., 2008). The sources, sinks and atmospheric concentrations of
reactive trace gases, greenhouse gases and aerosol particles depend strongly on each other via physical, chemical and biological
processes (e.g. Arneth et al., 2010, Stocker et al., 2013; Kulmala et al., 2014a; Unger, 2014, Green et al., 2017; Smith et al.,
2023). Furthermore, both human actions and natural feedback mechanisms between the biosphere and atmosphere have
substantial impacts on interactions between these atmospheric constituents and their influences on air quality and climate (Raes
et al., 2010; Stocker et al., 2013; Kulmala et al., 2015; Kulmala, 2015; Nolan et al., 2018; Doherty et al., 2022; Wang et al.,
2023). The importance of atmospheric aerosol particles on climate and human health in both regional and global scales has
attracted a plenty of research interest during the recent years (Butt et al., 2017; Boy et al., 2019; Bellouin et al., 2020;
Lappalainen et al., 2022; Lintunen et al., 2023). Despite these efforts, atmospheric aerosol particles remain perhaps the least
known factor influencing radiative forcing, causing thereby large uncertainties in predicting the future behavior of the climate
system (IPCC 2013, 2021).

The global annual cost of climate change impacts are estimated to reach hundreds of billions of euros by 2030 (UNEP, 2016;
Köberle et al., 2021) and, with an increasing global warming, this cost is expected to increase strongly in the future. According
to the World Economic Forum, climate change has cost the EU €145 billion in a decade
(https://www.weforum.org/agenda/2022/12/climate-europe-gdp-emissions/). There is an urgent need for improving climate
projections, reducing greenhouse gas emissions and developing options to sequester terrestrial carbon by simultaneously taking
into account the other climate forcers, such as atmospheric aerosol particles and trace gases. In addition, in order to have better
understandings of natural and anthropogenic sources and sinks of carbon and of atmospheric processes influencing air quality,
we need to develop existing monitoring and forecasting systems of the terrestrial carbon cycle, and to both enhance and
improve measurements from process levels to a global scale. These practical needs provide emerging business opportunities
for various industries. For example, European Green Deal Investment will mobilize at least one trillion euros of sustainable



investments over the next decade (COM 2020). The information produced by using new verification systems is essential for society to design economically and socially optimal sustainability strategies and climate-neutrality pathways and to be able to
meet Paris Agreement targets (Kriegler et al., 2018).

In order to meet Environmental Grand Challenges, comprehensive and open data are essential. We need data from international infrastructure networks, such as GAW (Global Atmospheric Watch), and from European environmental research infrastructures, such as ICOS (Integrated Carbon Observation System), ACTRIS (Aerosols, Clouds, and Trace gases Research
Infrastructure), eLTER (Integrated European Long-Term Ecosystem, Critical Zone & Socio-Ecological Research Infrastructure) and AnaEE (Infrastructure for Analysis and Experimentation on Ecosystems). To find answers to the interlinked Environmental Grand Challenges, we need integration of the different infrastructures dedicated to measure each of the Earth components. An example of such an approach is the top-level, interdisciplinary SMEAR (Station for Measuring Earth surface – Atmosphere Relations) network that produces knowledge of the interactions and feedbacks between the Earth components
(biosphere, hydrosphere, atmosphere, and geosphere), spearheading science-based solutions related to the interlinked Grand Challenges.

Here we present the SMEAR concept, different ways to utilize it and also show the potential future development in which *in situ* and remote sensing data can be used seamlessly with each other and with multiscale models.

**2 SMEAR concept**

The SMEAR concept is based on comprehensive, continuous, and integrated long-term observations (Hari and Kulmala, 2005). It has been developed to meet Environmental Grand Challenges and to collect big open data sets in order to test theories and develop models at the interfaces of different Earth components. Such unique environmental open data can contribute to solving burning questions of society – even questions that are currently unforeseen.

Current observations (see IPCC 2013, 2021) are fragmented, which means that typically different infrastructures are measuring greenhouse gases, aerosols, air quality, ecosystems, climate and biodiversity. These measurements are conducted in different locations and environments, and often during relatively short campaigns. However, to meet the ongoing environmental challenges, an integrated approach with long-term measurements is needed. In practice this means co-location of various
infrastructures, which enables simultaneous measurements of different Earth components (Kulmala et al., 2021, Lintunen et al., 2023). Changes in one of these components are directly or indirectly communicated to the others via intricately linked processes and feedbacks occurring at their interfaces (e.g. Stocker et al., 2013; Nolan et al., 2018; Smith et al., 2023).



The SMEAR stations together with the SMEAR concept were established before the various international environmental
research infrastructures were established (Hari and Kulmala, 2005). Today the four Finnish SMEAR stations and international
SMEAR-like stations (Kulmala et al., 2021) are contributing to several of them, namely ICOS, ACTRIS, eLTER, AnaEE,
WMO/GAW and/or EMEP.

Crucial in the SMEAR concept is that it measures atmosphere-Earth surface interactions and feedbacks. The Earth surface can
be forest, lake, peatland, urban area, glacier, agricultural land, etc. The most established station, SMEAR II, is located in
Hyytiälä, Finland, and it includes measurements of over 1200 different variables (Kulmala et al., 2021). The measurements
are conducted at different scales from small chamber enclosures to a regional scale, which is made possible by the 128-m-high
measurement tower at the SMEAR II station. The measurements include meteorological variables, atmospheric compositions
and fluxes (aerosols, clouds, atmospheric chemistry, greenhouse gases etc.), as well as variables describing ecosystem
functioning and soil dynamics. Long-term *in situ* measurements are accompanied by remote sensing, experiments (both lab
and field) and multi-scale modelling. Such an approach enables us to evaluate e.g. trends in measured concentrations and
fluxes, process dynamics, and feedbacks between processes and Earth components, such as soil-forest-atmosphere and forest-
soil-streams-lake.

An important part of the SMEAR concept is open access to the research infrastructure and open data
(https://smear.avaa.csc.fi/). These data are massive and heterogeneous, and thus challenging to manage, but the easy access to
these data and both harmonised and standardised ways to analyse it are important (Junninen et al., 2009). As a summary, we
can state that we need to work towards an open, integrated approach that can be accomplished with a global SMEAR network,
a global Earth observatory (Hari et al., 2016, Kulmala 2018).

In the following section, we give examples of capabilities of the SMEAR concept addressing different kinds of research
questions: atmospheric new particle formation, COBACC feedback loop that combines terrestrial carbon sink to aerosol
source, COVID impacts on air quality, and long-term trends in some of the quantities essential for air quality and climate
research. We also discuss the future role of comprehensive measurements to detect unexpected changes in the environment
and atmosphere.

## 3 The utilization of comprehensive data sets

Here we provide examples on how we have used comprehensive data sets to meet several scientific and societal challenges
and discuss briefly what has been learnt from these investigations.



### 3.1 Atmospheric new particle formation (NPF)

A well-known example of the usefulness of comprehensive, long-term observations are investigations related to atmospheric NPF (e.g., Mäkelä et al., 1997; Kulmala et al., 2013). Before such observations, more emphasis was placed on binary nucleation in the stratospheric conditions (e.g, Hamill et al., 1982). When looking at the scientific literature published prior to mid-1990´s, the common thought appears to have been that in the troposphere, NPF is a relatively rare and local phenomenon with minor contributions to regional or global aerosol particle budgets. However, the first long-term observations of particle number

concentrations revealed a frequent occurrence of NPF and its regional character in a boreal forest environment (Mäkelä et al., 1997), and later observations confirmed the same to be the case in many other types of atmospheric environments (e.g. Kerminen et al., 2018; Nieminen et al., 2018; Chu et al., 2019; Brean et al., 2023). Motivated by long-term observations, explicit description of NPF was then included in several large-scale modeling frameworks. Simulations using such models demonstrated that NPF is the dominant source of the particle number concentration in the global atmosphere, and an important

contributor to concentrations of cloud condensation nuclei (e.g. Merikanto et al., 2009; Gordon et al., 2017).

In order to understand how atmospheric NPF is connected with climate and air pollution, or emissions to the atmosphere, one needs to quantify the mechanistic pathways of atmospheric NPF and its basic characteristics, such as its frequency and associated particle formation and growth rates. Our mechanistic understanding on the initial steps of atmospheric NPF,

clustering, relied for a long time on theories and laboratory experiments, and with a belief that the only important clustering mechanisms in the atmosphere is the binary nucleation between sulfuric acid and water vapors (Malila, 2018). Atmospheric observations provided increasing evidences on the existence of multiple and possibly more complex clustering pathways (e.g. Kulmala et al., 2014b). Such findings inspired comprehensive and dedicated laboratory experiments in the CLOUD chamber at CERN (e.g. Kirkby et al., 2011; Almeida el al., 2013; Lehtipalo et al., 2018, He et al., 2021). Many of the clustering pathways

quantified in these CLOUD experiments have recently been found in atmospheric observations (e.g. Sipilä et al., 2016; Jokinen et al., 2018; Lehtipalo et al., 2018; Beck et al., 2021; Yan et al., 2021), confirming the diversity of NPF in various atmospheric environments.

Atmospheric observations have revealed large differences in the NPF characteristics between different sites, as well as between

different seasons at individual sites (e.g. Nieminen et al., 2018; Chu et al., 2019; Deng et al., 2020, Brean et al., 2023). At sites with multi-year observations, there appears to be a notable inter-annual variability in both frequency and intensity of NPF, and in some cases also a long-term trend has been reported (Asmi et al., 2011; Nieminen et al., 2014; Saha et al., 2018; Kalivitis et al., 2019; Neefjes et al., 2022). Figure 1 shows the monthly medians of nucleation mode particle concentrations measured at the SMEAR II station. Over the 27-year observation period from 1996 until 2022, the monthly median concentrations

decreased at a rate of –0.9%/year. The temporal variability in NPF characteristics has been ascribed to changes in meteorological conditions and aerosol precursor sources, including clear influences of reduced sulfur emissions and other air



pollution control actions in Europe and North America (Hamed et al., 2010; Kyrö et al., 2014; Wang et al., 2017; Saha et al., 2018) and more recently also in China (Zhao et al., 2021; Zhu et al., 2021).

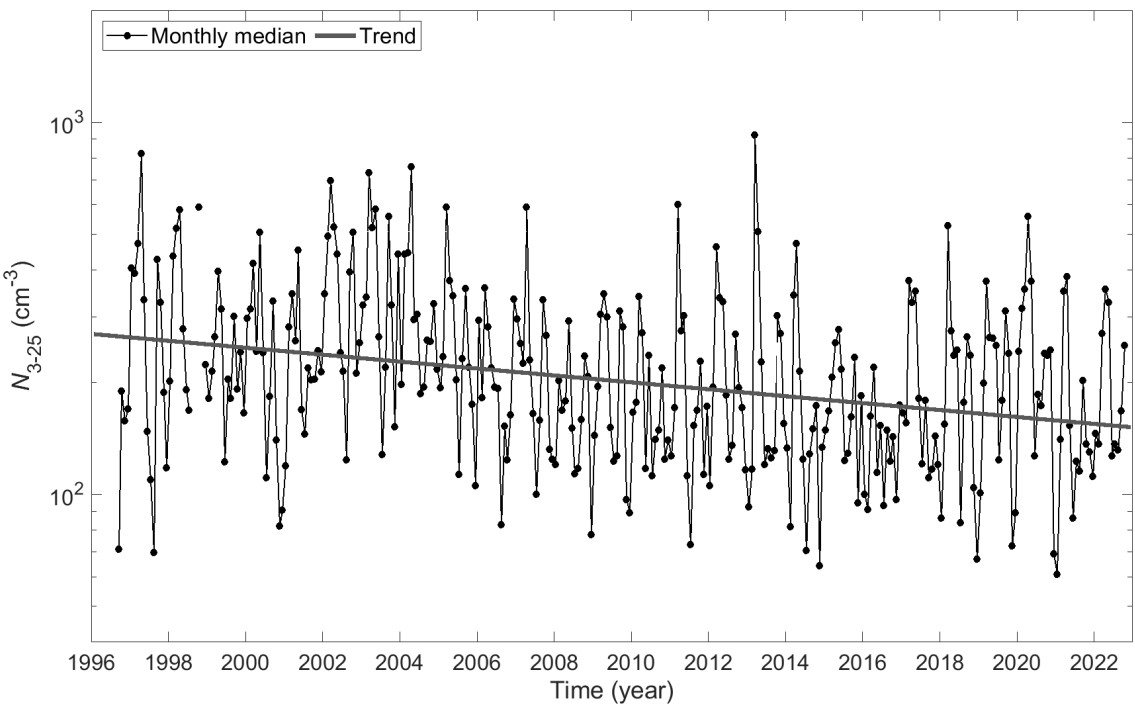


**Figure 1: Monthly medians of the nucleation mode particle (3–25 nm in the mobility diameter) concentrations at the SMEAR II station in Hyytiälä, Finland. The line is a linear lin-log fit to the data and shows clearly the decreasing trend.**


Long-term atmospheric observations have played a central role in atmospheric model development. The first semi-empirical parameterizations of new particle formation rates for modeling purposes were based on simultaneous and continuous measurements of gas-phase sulfuric acid concentrations and particle number size distributions (Sihto et al., 2006; Paasonen et al., 2010; Semeniuk and Dastoor, 2018). Later, long-term observations have been essential in testing the performance of large-
scale models in simulating NPF and subsequent growth of newly formed particles to CCN (e.g. Spracklen et al., 2010; Fountoukis et al., 2012; Yu et al., 2015; Qi et al., 2018).

While models are likely to be the main tool for estimating the future impacts of atmospheric NPF on climate and air quality, they regularly need observations to verify their performance. In addition, many related scientific issues remain that cannot be
solved without comprehensive and continuous observations. One of them is the relative importance of different clustering



pathways in different environments and due to continually changing atmospheric composition in these environments. The second issue is the quantification of factors dictating the frequency and intensity of NPF, including the role of "quiet NPF", i.e. relatively weak NPF not captured by traditional NPF event analysis methods (Kulmala et al., 2022a). The third issue is to understand the growth of newly formed particles into sizes where they may act as CCN or contribute to haze formation (e.g.

Ren et al., 2021; Kulmala et. 2022b; Stolzenburg et al., 2023). Related to this issue, we need long-term observations to better understand how small clusters survive while growing  larger sizes, especially in polluted environments (Kulmala et al., 2017; Tuovinen et al., 2022), in order to find out the relative importance of condensation and heterogeneous reactions in growth, and to quantify the most important precursor vapors causing this growth.

## 3.2  COBACC and other feedback loops

To understand better the complex feedbacks between the atmosphere and ecosystems, we have developed a concept called the Continental Biosphere-Atmosphere-Clouds-Climate (COBACC) feedback loop (Kulmala et al., 2013). It utilises a multidisciplinary-and integrated approach to quantify the feedbacks. The loop consists of several interrelated processes (Fig. 2, for detailed description see Kulmala et al., 2014a; Artaxo et al., 2022, Kulmala et al., 2023): 1) the atmospheric temperature and $CO_2$ influences on biogenic volatile organic compound (BVOC) emissions, 2) the influence of BVOCs on the formation

and growth of aerosol particles, 3) the effect of aerosol particles on clouds, 4) the effect of aerosol particles and clouds on solar radiation, in particular, on its diffuse fraction, and 5) the link between diffuse radiation and photosynthesis, and carbon sink in general.

The various processes involved in the interactions within the COBACC feedback loop occur over different time scales.

Increases in atmospheric temperatures and carbon dioxide concentrations occur at inter-annual time scales, carbon cycling including photosynthesis and emission of BVOCs vary on scales from sub-hourly to seasonal, while the cloud variability and its effect on radiation that drive photosynthesis operate at sub-hourly time scales. Therefore, continuous and comprehensive observations, together with modelling, are key to solving such complex questions. In what follows, we summarize our current understanding of the feedback loop in the boreal zone (for a more comprehensive review see Artaxo et al., 2022), and indicate

future directions. Continuous observations serve as a base for most of the studies cited below, and the SMEAR II data set, used in most of them, remains the most comprehensive one up to date.

Paasonen et al. (2013) considered the effect of warming climate on BVOC emissions and associated increases in >100 nm aerosol particle number concentrations (a proxy for CCN), quantifying the potential cooling effect due to this feedback. The

pilot study of Kulmala et al. (2014a) made the first estimate of the COBACC feedback loop using SMEAR II data, focusing on the direct aerosol effect and excluding clouds from the consideration. After that, Ezhova et al. (2018) refined and extended the analysis of the aerosol-diffuse radiation-photosynthesis part of the feedback loop using data from five sites in the boreal zone, also excluding clouds. Clouds were included in the Earth System Model (ESM) studies on the feedback loop (Rap et al.,



2018; Sporre et al., 2019). However, the link between BVOC, aerosol particles and clouds in various ESMs is a source of substantial discrepancies, even of different sign, in the radiation – the main driver of photosynthesis (Sporre et al., 2020). Therefore, observations remain an extremely relevant source of data for this complex question.

Based on the COBACC feedback loop, Kulmala et al. (2020) developed the CarbonSink+ concept, which, beside aerosols, takes into account the effect of forest on clouds and surface albedo. The next step is to include all radiative forcers. Current COBACC feedback loop studies are directed towards quantifying the role of clouds (Fig. 2), including their interaction with the surface-based parameters and their effects on radiation and photosynthesis, based on observations. The combination of on-site and satellite observations was employed to show that clouds become optically thicker in a warmer climate with larger amounts of organic aerosol particles (Yli-Juuti et al., 2021). Furthermore, Petäjä et al. (2022) showed that continuous interaction of an air mass with emissions from the boreal forest changes the properties of this air mass over a time period of several days, including both aerosol physical and chemical characteristics and humidity. Both factors are important for the formation and evolution of clouds. Räty et al. (2023) extended this approach to a data set covering more than a decade and confirmed the main conclusions. However, the outcome from this study regarding the effect of forest on cloud properties remains somewhat obscure. Cloud properties were taken from the satellite data sets, which drastically decreases the number of data available for analysis. To overcome this problem, e.g. the cloud classification algorithm by Ylivinkka et al. (2020) can be used. The algorithm allows quantifying some cloud properties, e.g. optical thickness for some types of clouds, whereas cloud fraction is linked to patchiness. The radiation measurements, an input parameter for this algorithm, have been measured at SMEAR II for more than two decades, and therefore the cloud-related data set can potentially be extended significantly. Overall, continuous comprehensive observations play a key role in tackling multidisciplinary problems with multiple time scales.

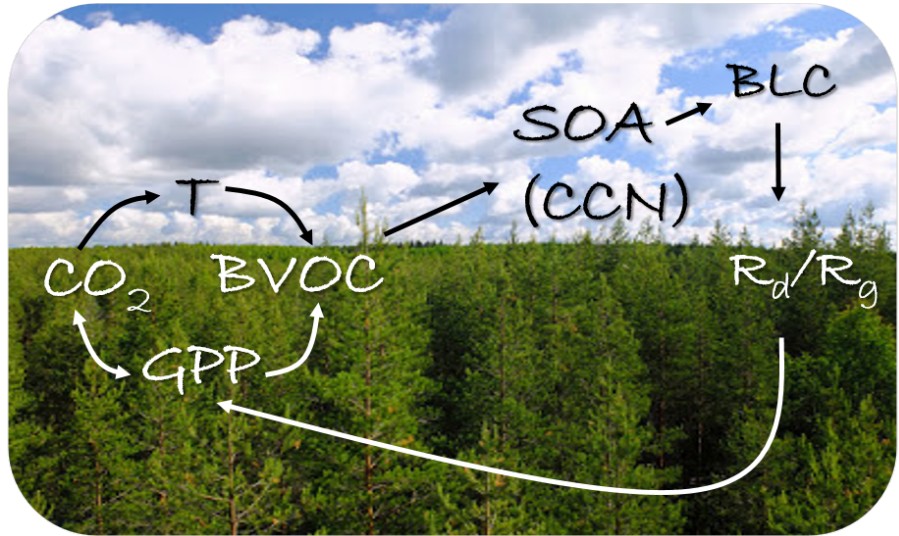






**Figure 2: Schematic of the COBACC feedback loop. T is temperature, BVOC – biogenic volatile organic compounds, SOA – secondary organic aerosol, CCN – cloud condensation nuclei, BLC - boundary layer clouds, Rd/Rg – diffuse fraction of solar radiation, GPP – gross primary production. Photo credentials ©Juha Aalto.**


While the COBACC feedback has, until now, been studied primarily in the boreal ecosystem, more data to constrain similar feedbacks within other ecosystems – particularly tropical and (semi)arid as well as urban – are urgently needed.

BVOC and semi volatile organic compounds (SVOC) are closely linked to SOA formation as a function of ecosystems.

Therefore, quantification and research of the fluxes of these VOCs is crucial. Field measurements of fluxes of BVOC and their oxidation products exhibiting reduced volatility, such as SVOC are challenging, and they can only be measured with rather short inlets to avoid wall losses during sampling. Recently a PTR3 instrument was used on top of the SMEAR II tower in Hyytiälä (Fischer et al., 2021) at 36 m above ground level and 15 m above the canopy of a forested ecosystem dominated by terpenoid emitters. The PTR3 instrument was installed approximately 4 m away from the tower structure and the virtually

wall-less inlet was successfully tested allowing undisturbed gas sampling from this distance. For the first time emission fluxes of sesquiterpene ozonolysis products and diterpenes were recorded. With the low flux signal-to-noise ratio achieved with the new instrumentation, we can now track and study clear diurnal patterns, even for the smallest emissions rates virtually in real time. Such intensive campaigns demonstrate the feasibility of new technology to be integrated in Flagship stations providing an extended parameter set in the future.


### 3.3 COVID restrictions

By the end of 2021, the global spread of COVID-19 caused by the SARS-CoV-2 virus has resulted in the loss of over 10 million lives (Adam, 2022; Msemburi et al., 2023). In China, national interventions were implemented starting from January 2020 to prevent the spread of the virus (China NHC, 2020). The strict lockdown measures associated with the COVID-19

pandemic provide a unique opportunity to investigate, in a real-world atmospheric laboratory, the direct and indirect effects of reduced emissions, as well as atmospheric chemistry and interacting processes associated with these emission changes, on air quality (e.g. Kroll et al., 2020; Jiang et al., 2021; Wang et al., 2021; Sokhi et al., 2021; Amouei Torkmahalleh et al., 2021).

This unique opportunity is a good demonstration of the strength of the SMEAR concept applied to atmospheric observations.

First, in such an unplanned situation where new research activities have also been restricted, it is impossible to organize and carry out targeted intensive observations. Second, although there are several functioning observing stations, the relatively poor measurement capacity in most of them is unable to support the in-depth analyses needed for new scientific insights. To date, there are hundreds of atmospheric science studies relevant to the COVID-19 lockdown (https://docs.google.com/document/d/1UTQvW_OytC37IatMNR5qJK7qKfSylNpI2fT3pdteVZA/edit). However, a large



proportion of these studies only report variations of a few atmospheric parameters and are far from providing a mechanistic understanding of changes in atmospheric processes. There are also studies that use regional models to understand the atmospheric processes during the lockdown, but these modeling results have limited verification due to the lack of comprehensive observations.

The Aerosol and Haze Laboratory of Beijing University of Chemical Technology (AHL/BUCT; Liu et al., 2020) is one of the stations fully implementing the SMEAR concept. This station was established in January 2018, and since then it has been operating uninterruptedly with a full measurement capacity. Our comprehensive observations showed that the lockdown caused changes of different magnitudes in various atmospheric parameters (Fig. 3). In general, most of the primary pollutants, such as NOx, SO2, BC, and VOCs, showed a reduction in their abundance, but at different levels. For example, NOx was

reduced by more than 50%, SO2 by ~25% and VOCs only by ~15%. This suggests that emissions from different source sectors were affected differently by the lockdown. In contrast to the primary pollutants, most of the secondary pollutants showed increased concentrations. Particulate nitrate, sulfate, ammonia, organics, and gas-phase highly oxygenated organic molecules (HOMs) increased by ~50–150%. This indicates that secondary pollution, i.e. the conversion of primary pollutants into secondary ones, became more efficient. This phenomenon is closely related to the increased oxidation capacity of the

atmosphere, as indicated by the increased concentrations of OH, $NO_3$ and $O_3$.

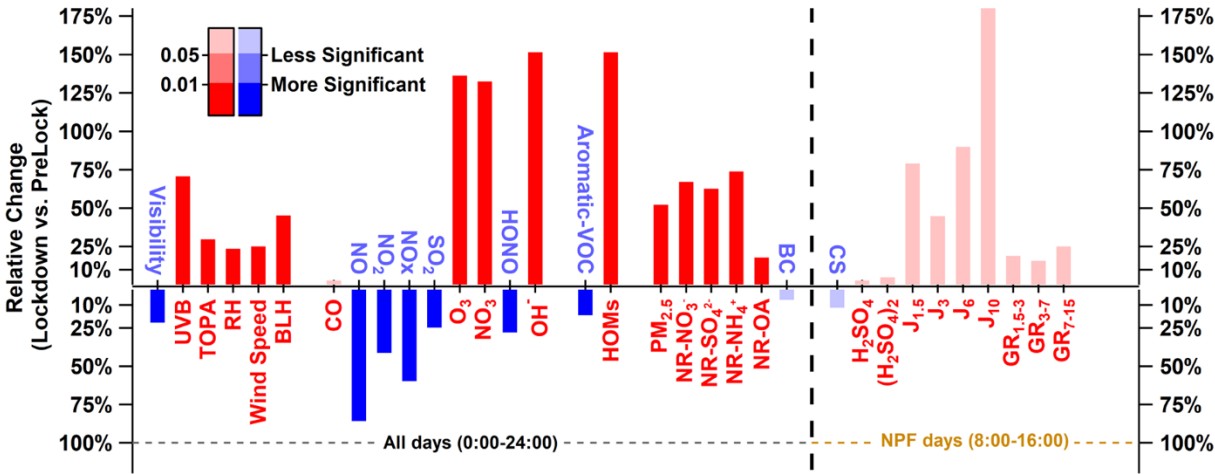

**Figure 3: Variations of primary and secondary pollutants caused by the lockdown. Relative changes of atmospheric variables**
**between the COVID-19 lockdown period ($24^{th}$ Jan – $5^{th}$ Mar 2020) and pre-lockdown period ($1^{st}$ Jan – $23^{rd}$ Jan 2020). The relative changes are defined as ([X]$_{lock}$-[X]$_{pre}$)/[X]$_{pre}$ × 100%, where [X] is the average of each variable. Variables associated with new particle formation (NPF) are shown only for NPF days during the daytime.**



Our comprehensive data sets allow us to obtain cutting-edge knowledge in several research directions. Here, we provide two
examples that provide direct observational evidence, showing the substantial influence of anthropogenic emissions on the
atmospheric oxidative capacity in both daytime and nighttime.

### 3.3.1 How did the atmospheric new particle formation responded to COVID-19 lockdown

Yan et al. (2022) explored how NPF responded to emissions reductions in Beijing during the COVID-19 lockdown. Clustering
between SA and base molecules drove the initial NPF in both the pre-lockdown and lockdown periods. Our results show that
this clustering was insensitive to emission reductions. Through direct observation, this study provided evidence that traffic
emissions do not appear to be a significant source of NPF in Beijing, in contrast to conclusions drawn from some recent urban
studies (Rönkkö et al., 2017; Guo et al., 2020).

During the lockdown period, we hypothesized that the reduction in nitrogen oxides ($NO_x$) concentrations would promote
particle growth. This is because NO can suppress particle growth by changing the composition of oxidized organic molecules
(OOMs) and making them more volatile on average (Yan et al., 2020). However, our study found otherwise. Although we
noted changes in the composition of OOMs, especially in molecules arising from the oxidation of aromatic volatile organic
compounds, there were only negligible changes in the volatility of OOMs. These results indicate that the reaction between
$RO_2$ and NO still plays a vital role in OOM formation even after a dramatic reduction in $NO_x$ levels. It has been suggested that
the autoxidation of $RO_2$ will become more important in atmospheric chemistry as $NO_x$ concentrations continue to decrease in
North America (Praske et al., 2018), leading to increased toxicity of peroxide-driven particles and the formation of secondary
organic aerosols (Zhao et al., 2017). However, our findings suggest that these harmful effects on human health and air quality
in Beijing are less likely to arise in the immediate future.


### 3.3.2 Enhanced formation of secondary organic carbon associated with $NO_3$ radical

Carbonaceous aerosols are acknowledged to have have significant impacts on climate change, Earth's radiation balance,
visibility, and human health (Donahue et al., 2009; Bond et al., 2013; IPCC, 2021). We examined carbonaceous aerosols
measured with an OC/EC analyzer between 1 December 2019 and 15 March 2020 in Beijing, encompassing the COVID-19
pandemic period (Feng et al., 2022). Our findings showed that anthropogenic gas-phase pollutants and primary organic
compounds were greatly reduced during the lockdown period. However, we also observed the emergence of enhanced
nighttime secondary organic carbon, which we attributed to nocturnal chemistry associated with the oxidation by $NO_3$ radical.
Our results indicate that this nocturnal chemistry phenomenon warrants greater attention in efforts to reduce PM concentration
in China.




## 3.4 Long-term trends in comprehensive observations of atmospheric variables

The long-term observations at the SMEAR II station in Hyytiälä, Finland, cover measurements of trace gas concentrations ($SO_2$, $O_3$, $NO_x$, CO, $CO_2$) as well as volatile organic compounds (VOCs, such as monoterpenes), which are measured on multiple heights above the ground at the 128-m-high mast. The continuous time-series of measurements starting from 1996
allow us to quantify long-term trends in these variables (Fig. 4). In addition, mass spectrometer measurements of sulphuric acid ($H_2SO_4$, the main oxidation product of $SO_2$) started in 2016, and proxy calculations based on the measured $H_2SO_4$ concentrations enable extending this time series (Petäjä et al., 2009; Dada et al., 2020).

The monthly-median concentration of the $H_2SO_4$ proxy has a decreasing trend of –2.6%/year (Fig. 4a). The $H_2SO_4$ proxy is
calculated based on the production of $H_2SO_4$ due to the oxidation of $SO_2$ by OH radicals and via stabilized Criegee intermediates which are produced in ozonolysis of monoterpenes (Dada et al., 2020), and on the loss of $H_2SO_4$ due to its condensation into pre-existing particles. Both the source and sink terms of $H_2SO_4$ have decreasing trends in Hyytiälä during 1996–2022, being –2.7%/year for the $SO_2$ concentration –1.0%/year for the condensation sink (Fig. 4c-d). The stronger decrease in the $H_2SO_4$ precursor vapor concentration compared with the $H_2SO_4$ sink seems to determine the long-term trend
observed in the sulphuric acid proxy concentrations. The monoterpene concentrations are characterized by a larger year-to-year variability and do not show a statistically significant trend (Fig. 4b). In the summertime, the monoterpene concentrations are highest during the year and have stayed relatively constant, whereas the annually lowest concentrations during winter and spring show also the largest variability between years. This emphasizes the importance of versatile, comprehensive measurements for quantifying atmospheric processes.



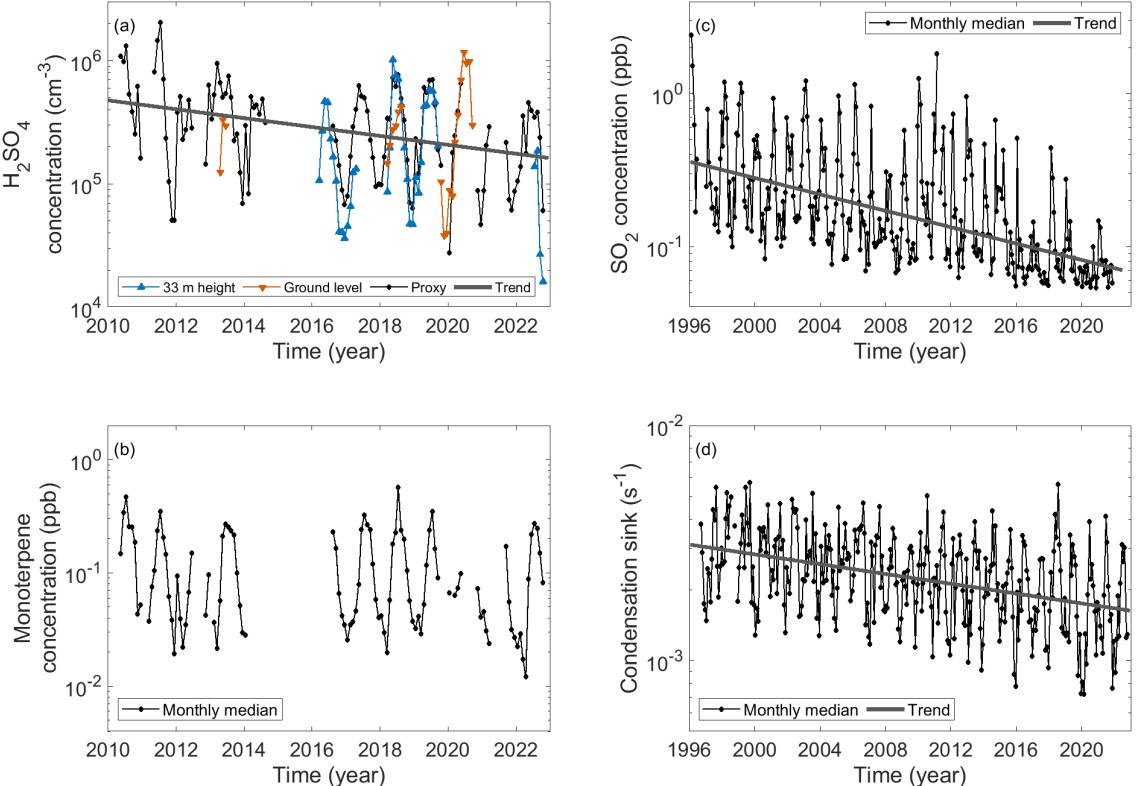

**Figure 4: Trends in (a) sulphuric acid concentration, (b) monoterpene concentration, (c) sulphur dioxide concentration, and (d) condensation sink. Note the different time periods: in panels (a) and (b) 2010–2022, and in panels (c) and (d) 1996–2022. The dotted black lines show monthly medians of observations and the grey solid lines are trends fitted to the logarithmic values of the monthly data. Monoterpene concentrations do not have a statistically significant trend and therefore the trend line is not shown in panel (b).**



## 4 Integration of the data from satellites, models and in situ observations

### 4.1 Satellite and Airborne observation

Satellites provide data on a global scale of atmospheric composition, radiation, surface properties and meteorology. Passive satellite measurements of atmospheric gases and aerosols are representative over an entire atmospheric column, and hence they are not directly comparable to in situ measurements. Although satellites cannot provide as detailed and wide range of different atmospheric parameters as comprehensively equipped in-situ measurement stations, such as SMEAR II, with their spatial coverage they can provide very valuable and complementary information. After understanding and analyzing in-situ and satellite measurements together on a station-by-station basis, satellite data can enable the transition from point-like measurements to the interpretation of regional variability in atmospheric processes (e.g. Viatte et al., 2021; Pseftogkas et al., 2022; Hakala et al., 2019).





One example of utilizing satellite data, when moving from pointwise to global (or regional) analysis, has been to better understand the new particle formation (NPF) phenomena on a larger spatial scale (Kulmala et al., 2011; Sundström et al., 2015). As satellites cannot essentially detect aerosol particles smaller than 100 nm in diameter, these observations as such are

not directly applicable for NPF studies. However, satellites provide information on many atmospheric parameters tightly linked to NPF, such as UV radiation, trace gas concentrations and estimations on ambient aerosol loads. By analyzing these data with detailed in situ measurements, it is possible to develop merged satellite variables that can be used to study the regional variation of NPF. Other examples are various climate-related feedback loops, e.g. between the atmosphere and the biosphere, where satellite observations could have the potential of increasing the understanding of the spatial scale variation. Such development

work would not be possible without SMEAR-type observations that have the capability of providing process-level understanding of the phenomena. It is also essential that such extensive in situ observations exist in various environments, so that the sensitivity of satellite observations in these kinds of applications could be properly tested.

Passive satellite instruments provide typically so-called columnar measures, and for instance aerosol optical depth (AOD) is

the vertically-integrated aerosol extinction. Similarly, gas concentrations measured by satellite represent column or partial column concentrations, typically over a tropospheric column. Therefore, comparison with surface in-situ measurements cannot offer a direct validation for the measurements made by satellite instruments. However, all possible columnar measurements (e.g. AOD by ground-based sun-photometers) at the same ground station, on the other hand, would facilitate satellite data validation and provide information on the accuracy of the satellite observations. Currently, the SMEAR II station in Hyytiälä

is accompanied with an Aerosol Robotic Network station (AERONET, Holben et al., 1998), which allows a direct validation of satellite-based aerosol observations. Moreover, satellite validation would strongly benefit from reliable gas and aerosol vertical profile observations from the surface level up to the stratosphere.

One possible future pathway in better bridging the spatial-scale gap between pointwise in-situ measurements and large-scale

satellite measurements would be to utilize unmanned aerial vehicle (UAV) measurements (Motlagh et al., 2023). Satellite data have spatial resolutions limited to a few hundred meters at best, and in atmospheric observations more typically to kilometers. Drone measurements (e.g., Kezoudi et al., 2021) could bring more insight into the sub-kilometer scale variations if carried out in the vicinity of SMEAR-type stations at the time of the satellite overpass.

## 4.2 Model frameworks

Model frameworks for the global climate and Earth systems have been constructed to replicate real-world processes and interactions as closely as necessary to understand the current state of these systems, reasons for past changes, and ultimately to simulate future climate pathways in order to support adaptation and mitigation efforts (Bauer et al., 2021). Modern Eerth System Models (ESMs) combine an increasing number of individual components, including not only the physical ocean and



atmosphere models but also detailed descriptions of chemistry, aerosols and the biosphere (e.g. Döscher et al., 2022). This
increase in model complexity has established groundbreaking research of Earth system feedbacks and quantification of their
strength in current and future climate (Sporre et al., 2019; Thornhill et al., 2021).

Despite rigorous validation of its individual process components, evaluation and constraining of highly-coupled ESMs remains
difficult due to the large number of interactions and feedbacks within the Earth system (e.g. Sporre et al., 2020). In the temporal
and spatial scales of ESMs even the observational record as a whole remains brief and irregular, and therefore the observations
must be extended by proxies of changes in the historical period (Wandji Nyamsi et al., 2020) and towards the Earth's deep
past over millions of years (Wong et al., 2021). Only integrated long-term observations can provide multidisciplinary data to
support evaluation of simulated Earth system feedbacks and their components.

With increasing complexity and process details, ESMs can arrive at correct results via wrong reasons and counteracting biases.
The advancement of spatial resolution and process descriptions within ESMs already allows evaluation at the process-scale.
For example, NPF events can be co-analyzed from ESMs and long-term datasets (Bergman et al., 2022). Such process-oriented
analysis is essential for validating the reasons for biases or systematic errors in simulated properties (e.g. CCN), but this
requires dedicated long-term observations to constrain the models throughout distinct climate states and changing environment
(Fanourgakis et al., 2019). Recent advances in trajectory-based analysis of ESMs provide a novel way to investigate simulated
air-masses and station footprints co-located with observations. To complete a global 4D evaluation of ESM performance, the
long-term stationary datasets should be complemented with surface or airborne transects, vertical profiles and satellite
retrievals (e.g. van Noije et al., 2021).

With the competition between increasing spatial resolution and more detailed process descriptions in ESMs, data-driven
approaches have been suggested to replace computationally intensive modules (Ahola et al., 2022). Whether through
emulation, neural networks or other machine learning techniques, the teaching and learning process requires comprehensive
understanding of the model realm complemented with integrated observational suite (e.g. Schreck et al., 2022).

**4.3 Integration of different approaches**




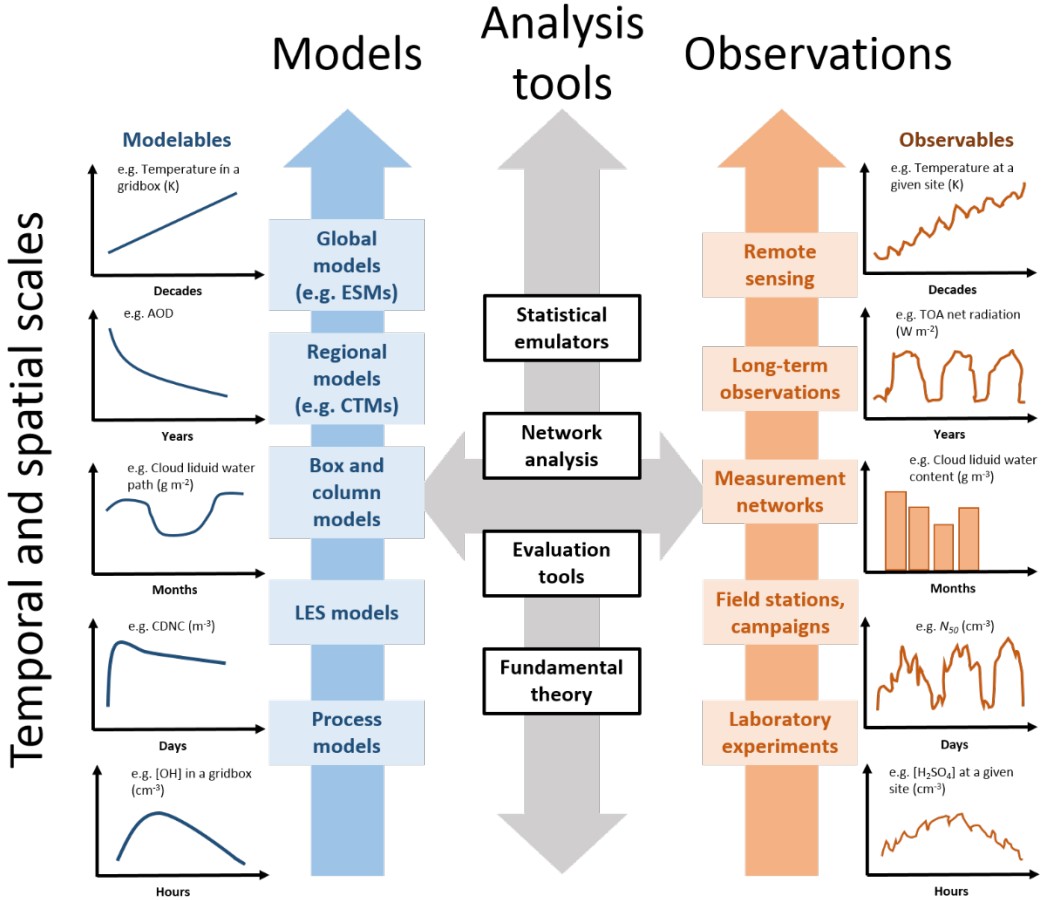

**Figure 5: Combination of methods to integrate "bottom-up" and "top-down" insights on atmospheric aerosol and its interactions with clouds, as outlined within the FORCeS project (see forces-project.eu. Figure courtesy of Tinja Olenius.**

Combination of emerging long-term in-situ measurements, satellite data and process understanding bear a great potential for finding new ways to evaluate and constrain ESMs, and to reduce uncertainties in their projections (see Fig. 5, adopted from the FORCeS project). The inevitable spatial limitation related to in-situ observations can, to some degree, be overcome by ensuring long enough temporal coverages and enhancing the number of representative data points (to be compared with satellites and models) this way (e.g. Isokääntä et al., 2022; Khadir et al., 2023).

Long-term, global in-situ observations are specifically useful in pin-pointing the model weaknesses and strengths, as well as in providing detailed observations with various techniques at well-defined altitudes, as opposed to sampling the entire atmospheric column. Long-term in-situ observations offer great opportunities to compare detailed process-level observations with satellite observations and large-scale models. Detailed measurements enable e.g. investigations of size-segregated trends in aerosol loadings in a regional context using both ESM and observational data (e.g. Leinonen et al., 2022). While the number



of relevant data is steadily increasing, more long-term observations from under-sampled parts of the atmosphere (global south, highly remote areas) are needed. In addition, combining long-term in-situ measurements, satellite data, ESM model outputs and process understanding offers a great potential for finding new ways to evaluate and constrain the biosphere-climate feedbacks in ESMs, the magnitude of which is still highly variable between models (Thornhill et al., 2021). By separating the different processes (e.g. those that relate biogenic emissions and resulting aerosol concentrations to air temperatures and cloud

properties), and by combining long-term observations with satellite data and multiscale modelling, one can facilitate the evaluation of the predictive abilities of the models (Blichner et al., to be submitted). This approach can help to isolate the impact of individual factors and improve our understanding of the underlying processes.

For example, the uncertainty in the effective radiative forcing due to aerosol-cloud interactions is governed by the cloud

susceptibility to aerosol perturbations (Bellouin et al., 2020). This is split into two components which are (i) the response of the cloud droplet number concentration to aerosol perturbations – relevant for the radiative forcing due to aerosol-cloud interactions, also known as the Twomey effect – and (ii) the rapid adjustments in particular of cloud liquid water path and cloud fraction (Bellouin et al., 2020). In-situ long term aerosol and cloud observations enable investigations of cloud-susceptibility to aerosol perturbations. In-situ observations (both long-term and campaign-wise) and process

understanding combined with ESM model outputs (or with satellites), facilitate pin-pointing specific processes or factors that should be improved in order to be able to describe the cloud activation and aerosol indirect forcing correctly in the models.

## 5 Future perspectives and possibilities

Currently, the speed of climate change along with its unpredictable consequences are challenging the capacities of existing observation systems. In addition, the ability to analyze the yet unknown questions and challenges, the "black swans" (Taleb,

2010), calls for comprehensive continuous observations. For example, COVID19 gave an unexpected opportunity to demonstrate the effect of exceptional reductions in anthropogenic emissions on air quality and climate (e.g. Gettelman et al., 2020; Wijnands et al., 2022). In this case the already running SMEAR-type, comprehensive measurements at the BUCT/AHL station in Beijing enabled us to investigate the atmospheric processes in detail. Other examples of this type of unusual and extraordinary events could be volcanic eruptions, gas pipe attacks, extreme weather events, forest fires, exceptionally dry

periods, economic collapses or chemical weapons. All these events have both short- and long-term dynamic effects on air quality and the climate system as well as on the functioning of societies. Also, the possibility of realized global tipping points, such as permafrost loss or boreal forest shift towards tundra, may lead to unexpected environmental episodes, events and feedbacks (Rockström et al., 2009, Kulmala et al., 2015, Lenton et al., 2019). The key questions are whether the majority of current observation systems contain sufficiently comprehensive set of variables to capture these events, and whether we have

the preparedness to detect, analyze and quantify these events.



Recently proposed geoengineering approaches for mitigating global warming include a clear potential for large magnitude feedbacks which can have significant, yet unpredictable consequences to other processes. Risks related to uncontrolled geoengineering without international laws and the manipulation of the atmosphere highlight the value of continuous,

comprehensive measurements detecting the changes. For example, operational Solar Radiation Modification (SRM) deployment would introduce new environmental and socio-economic threats like damaging the ozone layer and overcompensating climate change at regional scales (UNEP, 2023).

We need open big data to meet the present grand challenges, and we need to collect comprehensive data to be able to answer

questions which do not exist today. The questions can be societal, economic or scientific, or any combination of these. To effectively collect, distribute and utilize big data, there are several key actions that needs to be considered:

Firstly, it is important to promote open data flows and storages globally via open access data platforms and structures. This can be achieved through optimizing data flows by also considering how to access and analyze the data at the storage site instead

of transferring huge amounts of data. Advanced AI and data mining techniques should be employed to explore and utilize the data effectively. It is also important to transfer knowledge to make the data more accessible and to develop and provide examples and roadmaps for local data owners that highlights how to get merits via open data. The more local the data and data needs are, the more challenging it is to have them fully open. Therefore, it is crucial to demonstrate the benefits of offering open access to the data.


Secondly, we need to develop measurement protocols and data standards to reliably observe concentrations, fluxes and changes in the atmosphere and the environment. When using low-cost sensors, it is crucial to establish a proper calibration system that ensures data quality and traceability. Also, existing observation station types require calibrated sensors and enhanced harmonization. It is important to connect to existing harmonization actions by international organizations, such as the European

Committee for Standardization (CEN), European environmental Research infrastructures and Network of Air Quality Reference Laboratories (AQUILA) etc. Several World Meteorological Organization working groups are already active towards these goals. However, we need to go to the next level to make in-house processes more effective.

Finally, it is necessary to establish a hierarchy of stations ranging from cost-effective sensors (low cost) to comprehensive

flagship stations, such as SMEAR – Stations to Measure Earth surface Atmosphere Relationships, by utilizing the knowledge and experience from the European Strategy Forum on Research Infrastructures (ESFRI) as well as from operational observation networks pertinent to different domains in the atmosphere – environment continuum. Within the next 20 years, we should have a station network utilizing the hierarchy of stations with three steps, namely Flagship stations, Median stations and low cost sensors with enough Flagship stations scattered globally spatially and in ecosystem level to have enough representativeness of

varying conditions. The comprehensive Flagship stations should be preferably part of GAW network with 500–1000 stations



like SORPES, SIOS and SMEAR. The Median stations are high-end stations, but typically focusing on a specific topic (e.g. flux stations, AQ networks). Low cost sensors need calibration from Flagship and Median stations and utilizing of AI and 5G/6G/7G networks (Rebeiro-Hargrave et al., 2021).

Taking these actions will improve our understanding of the environment and our ability to respond to environmental challenges. An important question is who is willing to take the lead? Probably large international organizations are needed like WMO. In practice, we need to develop steps towards a GAW+ and maybe even to establish International climate /atmosphere institute. The institute should be multinational and multi-institutional research center following e.g. the model of CERN, but focusing on the atmospheric and Earth system research.


This would be based on combining the experiences from WMO Global Atmospheric Watch (GAW) program (WMO 2017), COPERNICUS, and international research infrastructures like the ESFRIs ICOS, ACTRIS and eLTER (see Section 1). The observation systems and research infrastructures, present standards, protocols and recommendations are consensus-based. For example, essential atmospheric variables and data products management alone have been developed by several different actors,

such as the Global Climate Observing System (GCOS) (WMO 2022) and GAW program (WMO 2017). These standardized systems have taken years to develop and are still in progress but need to be continued.

Under the WMO leadership, we should aim to the establishment of the global observatory for comprehensive data set(s) on *weather, climate, water and environment.* This framework would provide a wide range of benefits, such as creating a real-

world component and comparison for digital twin(s). It will also allow a proper WMO contribution to share integrated big datasets. The global observatory will provide a seamless connection between in-situ observations, remote sensing and multiscale model data. This will enable easy access and utilization of remote sensing products, such as inland water altimetry for rivers, lakes and reservoirs as well as arctic snow and ice cover. Furthermore, it would provide observational support for global food forecasting through real-time dissemination of river level and discharge data, air quality forecasting and food and

water supply forecasting. It would also enable us to predict future climate and find out feedbacks and interactions between various environmental factors.

Once we have collected all these data, it is crucial that it is utilized effectively. To be able to use the big data, open access is typically needed. However, there are several barriers before the data can be used. The barriers include the lack of

documentation, unknown data, misunderstood user needs, discipline specific jargon, bad and unusable interfaces, authorization problems, wrong terminology, training problems, unknown formats, difficulties in licensing and documentation, etc. To overcome the barriers of information, we need to have mutual trust and understanding of the needs, in addition to whcih we need to have access to the data to make new discoveries. This can be achieved by implementing the FAIR principles (Findable,

Accessible, Interoperable, Reusable) (Wilkinson et al., 2016), open data policies, proper knowledge transfer and conducting
impact investigations e.g. IIASA.

It is worth noting that the most important reason to investigate multiple variables with continuous measurements is that we never know beforehand, when we will meet a 'black swan'. When we have comprehensive, continuous, open data, we can analyze the data to study unexpected phenomena and answer to the upcoming challenges.

**6 Conclusions**

The need for comprehensive open data sets is obvious. Within the next 20 years, a new station network is needed, and the sooner the better.

Traditionally, and even in many cases today, there exist distinct infrastructures and their designated users, with experiences
rather far from each other. Different research groups have typically their own instruments, raw data, data analysis methods and publications. This is not an efficient way to meet grand challenges that can only be tackled with interdisciplinary approach. In the future, we need to utilize more joint efforts, including co-location of research infrastructures. Already to store raw data jointly, and to analyze it together in systematic ways, provides a big surplus. To have common data repositories for storing analyzed and published data is a big step forward.


One example on how comprehensive observations can be utilized is to solve air quality issues. In order to be able to understand the chemistry of air pollution, we need to observe multiple pollutants in existing air pollution cocktails (Kulmala, 2015). We should also remember that we spend 90% of our time indoors, and therefore also indoor air quality need to be understood. Air quality is important e.g. for health effects, visibility and from the acidification point of view. The multiple pollutants include
$PM_{2.5}$, size-resolved particle number, black carbon, $O_3$, $NO_X$, $SO_2$, CO, acids, various organic compounds etc., and their interactions and feedbacks. Our typical framework is a seamless chain from deep understanding to solutions, starting from observations and then continuing to understanding processes, feedbacks and interactions. These steps are needed to control air pollution and to improve air quality.

In basic and applied research, we make new discoveries and new knowledge from the resources we have with the money we have at our exposal. New discoveries and knowledge often lead to innovations and with innovations people make money from that knowledge. Innovations and new knowledge are a key for society to maintain and enhance wellbeing and this way the circle closes.



It is crucial to utilize multidimensional, multidisciplinary, multiscale approach to be able to answer questions related to grand challenges. It is also important to have a clear and ambitious vision from deep understanding to practical solutions. Also, we need seamless chain to connect measurements, modelling and theory as well as from research to innovations, economic growth and human wellbeing.

The main benefits that research community can gain from using integrated research infrastructure approach includes higher-quality science and higher visibility and recognition from the society and its various stakeholders and higher amount of scientific users both nationally and internationally. Moreover, collaboration possibilities will be enhanced, including feedbacks between domains, landscape analysis, up- and downscaling. Detailed experiments and observations can support each other, so that new observational and numerical methods can be developed. The improved utilization of data flows and synergies in data

use are foreseen.

In order to meet grand challenges and answer open scientific, societal and economic questions, we need to build upon a network of domain-specific research infrastructures, such as ACTRIS, eLTER and ICOS. We need to acknowledge that for example ACTRIS is already integrating several subfields, namely aerosol in-situ, trace gas and cloud in-situ observations and ground-

based remote sensing of aerosols, clouds and trace gases. The SMEAR concept in essence includes co-location and integration of the observations performed in the domain specific environmental RIs. A further connection to and integration with e.g., health and societal data are needed. Furthermore, we need excellent science, with high quality, critical mass and interdisciplinary research as well as education and training, i.e. knowledge exchange. We need to contribute to innovation ecosystem and have continuous, long-term dialogue with policy makers. Internationally, this enables clear contributions to

science diplomacy based on integrated scientific viewpoint.

### Acknowledgements

We acknowledge the following projects: ACCC Flagship funded by the Academy of Finland grant number 337549, Academy professorship funded by the Academy of Finland  (grant no. 302958), Academy of Finland projects no. 1325656, 311932, 334792, 316114, 325647, 325681, 347782, "Quantifying carbon sink, CarbonSink+ and their interaction with air quality"

INAR project funded by Jane and Aatos Erkko Foundation, "Gigacity" project funded by Wihuri foundation,  European Research Council (ERC) project ATM-GTP Contract No. 742206, and European Union via Non-CO2 Forcers and their Climate, Weather, Air Quality and Health Impacts (FOCI), and CRiceS (No 101003826). University of Helsinki support via ACTRIS-HY is acknowledged. Support of the technical and scientific staff in Hyytiälä and BUCT/AHL are acknowledged.




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
