# Peer review of "Opinion: The strength of long-term comprehensive observations to meet multiple grand challenges at different environments and in the atmosphere"

_EGUsphere, 2023_

## Author Comment (AC1)

Reviewer 1

Review of the manuscript "Opinion: The strength of long-term comprehensive observations to meet multiple grand challenges at different environments and in the atmosphere" from Markku Kulmala et al.

Indeed, the manuscript deals with a relevant issue, and the author's team is excellent. It deals with the need for a comprehensive global network of observations for the Grand Challenges. The manuscript describes the concept of SMEAR (Station for Measuring Earth surface – Atmosphere Relations).

**We thank the reviewer for the overall positive attitude toward our paper. We are confident that after revising the paper as described below in our answers, we will be able to address the major criticism by this reviewer and at the same time will improve the overall quality of our paper.**

The SMEAR concept is not new, and several papers from the same group have already discussed the same idea. Section 2 describes the concept and refers to the paper from Hari and Kulmala of 2005, which had already presented a very similar concept. The title of that paper is Station for Measuring Ecosystem– Atmosphere Relations (SMEAR II). So, what new ideas arise after 18 years from the same concept in this new manuscript? Other than the example of COVID and progress on NPF, the concept is basically the same.

**It appears that we have failed in communicating the main purpose of the current paper, at least partly, which explains the major criticism by this reviewer. It is very true that the SMEAR concept builds on research and ideas started by our community some time ago (actually more than 2 decades ago). However, due to the lack of long-term comprehensive observations, it has been challenging to test, or evaluate, the overall performance of the SMEAR II concept. Now, we are starting to have long enough data sets to do this. Therefore, the primary aim of the current manuscript is, by using a few examples based on measurements conducted mainly at the SMEAR II station, to demonstrate the performance of the SMEAR concept in addressing some of the global issues ("Grand Challenges") mentioned in our paper. We do not aim to provide a review on the state of knowledge on Grand Challenges, nor on current measurement networks related to these challenges. We admit that we have poorly communicated this in our submitted manuscript. We will revise the abstract of our paper and add some text to the end of section 1 to bring up our objectives more explicitly.**

The manuscript frequently discusses the "open DATA" issue when the discussion should be "Open SCIENCE." This indicates a quite narrow view since only having an open data policy does not promote open science.

**This is an important point. We surely support open science on top of open data, but this was probably not written clearly enough in the submitted manuscript. We will revise the text accordingly.**

Additionally, the manuscript proposes a **global** network of aerosol and other environmental observations. The GLOBAL is essential to the proposition. But 95% of the paper deals only with the approach adopted in one single monitoring station (Hyytiälä) in Finland. It almost exclusively discusses a single, very specific monitoring station. Not even other Scandinavian stations are mentioned. There is no discussion of global networks of stations such as GAW, NOAA, and others. Several examples globally are pretty similar to the SMEAR II station approach in Finland. I think this is an important limitation in the manuscript.

**Here, we return to our earlier answer on the main objectives of our current paper. We would, however, like to bring up that the SMEAR II station is not just a single measurement stations: it is probably the most versatile existing station with comprehensive long-term measurement data covering not only the atmosphere but also the biosphere and many interactions between these two. With these things in mind, it is understandable that we apply mainly papers relying on SMEAR II measurements when illustrating the performance of the SMEAR II concept. We agree that our list of other long-term measurement networks in section 1 and elsewhere in the paper is relatively short and probably too European centered. We will mention a few more relevant measurement networks in the revised manuscript, and add references outside those focused on SMEAR II measurements.**

Later, the manuscript describes the COBACC feedback loop, which is discussed only in terms of boreal forests and only with data from Finland. The manuscript would be much richer and more useful if the discussion on COBACC is done more broadly.

**We fully agree that estimating the strength of the COBACC feedback loop should be extended to temperate and tropical forests, and perhaps even to other continental biomes. We will mention this in the revised manuscript. There is at least one global modeling study on the COBACC feedback in a global atmosphere but, unfortunately, current observations are too limited to make any estimates on this feedback loop outside the boreal forest zone based solely on measurement data.**

Then on page 9, the manuscript discusses the COVID restrictions and the strength of the SMEAR concept. Again, it takes the example of the AHL/BUCT laboratory, where the first author also works. Furthermore, the authors choose a limited view of an important issue instead of a broader, European, or Global approach. This discussion would be much richer if done with a more comprehensive view.

**The AHL/BUCT measurements provide one example case in our demonstration of the performance of the SMEAR concept, similar to the other example cases considered in our paper. There are already several published reviews on different (regional/global) aspects of COVID restrictions. Moving our discussion into that direction would, on one hand, be a tremendous task that would lengthen and imbalance our paper considerable and, on the other hand, would be almost useless because of the abundant work published so far on this topic (see also our text on the first paragraph in section 3.3 referring to earlier work).**

The discussion that NPF is a very important issue in a proposed global monitoring network? What about aerosol radiative forcing changes? What about the aerosol optical properties linked to global heating/cooling? Certainly, many other aspects than NPF could have been discussed in the SMEAR and COBACC concepts.

**Aerosol optical properties and radiative forcing are certainly among the most relevant issues associated with atmospheric aerosols, and with atmospheric composition in general. We will acknowledge this in our revised manuscript. While we have made research on aerosol forcing estimates and optical properties based on measurements at the SMEAR II station, the associated time series are not among the longest ones discussed so far in the scientific literature (involving data from other stations/networks). As a result, these two issues were not selected for example cases to demonstrate the performance of the SMEAR II concept in this paper.**

Section 5 on "future perspectives and possibilities" also fell short of a global view. It would be important to analyze the GAW network, the European initiatives such as ACTRIS, ICOS, and the overall COPERNICUS. They are good examples, of course, and the manuscript could be more useful for readers if a critical analysis with recommendations for these initiatives could be included in Section 5. The need to have global networks for climate change monitoring as well as detailed atmospheric monitoring, including air pollutants, could be really good for the manuscript.

**The aim of our opinion paper was not to review global research infrastructure networks in general, but we agree that it is a good idea to clarify the additional need for an integrated, SMEAR-like station network. The main need is the need for comprehensive measurements of the different Earth components measured at same location, thus enabling multidisciplinary research on the interactions and feedbacks between them. Such data are needed for solving the grand challenges that are interlinked. Most existing research infrastructures are focused on certain scientific domain and/or Earth component, such as greenhouse gases, oceans, cryosphere, ecosystem processes or aerosols and clouds, and thus are not optimal for studying the interlinkages and feedbacks between the different Earth components. The SMEAR-approach fills this need. We will bring this up more clearly in the discussion in the revised version of the manuscript.**

Also important in the manuscript is the list of references. Of the 120 references, nearly 95% are from the same author, Kulmala. The manuscript could again profit from a broader list of European and American authors that have extensively worked on the paper subject. The proposal to have a global atmospheric observation network should not be done with a Finnish approach but certainly with a European or global perspective. The manuscript would profit significantly if thought much broader. The Global Challenges can only be fulfilled with a Global view.

**The main reason for having a big fraction of the cited studies focusing on measurements made at the SMEAR II station is that the primary objective of this paper, as stated in our earlier replies on these comments, was to demonstrate the performance of the SMEAR concept using a few example cases, not to provide a general review on the benefits of long-term measurements. However, we partly understand the critics by the reviewer in this respect, so we will add references outside those focused on SMEAR II measurements into the revised paper.**

Reviewer 2

This paper provides a discussion of the role of long-term environmental monitoring in understanding global environmental challenges. The value of long-term measurements in understanding environmental change still needs to be better recognised and this paper helps support this important argument.

The paper is well written and discusses a number of important topics. The focus is on one specific approach to long-term monitoring, namely the SMEAR (Station for Measuring Earth surface – Atmosphere Relations) concept. This focus results in the paper being heavily weighted to reviewing papers written by the lead author and co-authors, who lead the SMEAR approach.

The section on the impacts of COVID lockdowns on atmospheric composition is very interesting and clearly demonstrates the value of long-term measurements in understanding complex and unexpected environmental change. In particular, Figure 3 very clearly shows the value and need of long-term measurements across a very wide range of atmospheric components to help interpret and understand complex atmospheric and environmental interactions.

**We thank the reviewer for very positive comments, addressing to which will improve further out paper.**

I have a few comments that I would ask the authors to consider.

Major comments

The lead author and co-authors are productive researchers who have written a number of important review and opinion articles that cover similar ground to this paper. It would be helpful if the authors clearly identified how this article builds upon and is different to previous articles. In particular, it would be very helpful if the authors added a clear statement of the novel findings / focus of this study compared to previous papers.

**This is a relevant comment which, similar to the comments by the other reviewer, shows that we have failed in communicating the main purpose of the current paper. We base this paper on the SMEAR concept that builds on research and ideas that started in our community already more than 2 decades ago. The primary aim of the current manuscript is, by using a few examples based on measurements conducted mainly at the SMEAR II station, to demonstrate the performance of the SMEAR concept in addressing some of the global issues ("Grand Challenges") mentioned in our paper. We admit that we have poorly communicated this in our submitted manuscript. We will revise the abstract of our paper and add some text to the end of section 1 to bring up our objectives more explicitly.**

The paper focuses heavily on research conducted by the lead author and co-authors. This is understandable given the focus of the paper on the SMEAR concept established and run by the authors. Despite this, where relevant a wider acknowledgement of the work done by other groups, particularly other long term atmospheric monitoring, would strengthen the paper.

**We fully agree with this comment. We will add references outside those focused on SMEAR II measurements into the revised paper wherever appropriate/necessary.**

Minor comments

Page 7, Section 3.2. Figure 3 is a powerful way of illustrating the power of multiple long term observations in understanding a complex environmental issue. I missed an equivalent figure for the role of long-term measurements in helping to understand the COBACC and other environmental feedback loops. Is there an equivalent figure that demonstrates how measurements from SMEAR can be combined to help understandd a component of COBACC? If so, I think this would make a valuable addition to the paper.

**We will change the figure in Sec. 3.2 to illustrate boreal forest interaction with atmosphere (forest-boundary clouds link) based on results by Räty et al. (2023). This data set features 11 growing seasons from consecutive years of measurements at SMEAR II. The figure clearly demonstrates an increase in air mass fraction with higher, above median, values of cloud condensation nuclei, specific humidity and cloud optical thickness, as well as an increase in precipitation frequency.**

[Figure]

**Figure 2: Illustration of forest-boundary layer clouds link (Räty et al., 2023): fraction of air masses with the parameter value above its median after 10-30 h and 50-75 h interaction with boreal forest. Parameters are cloud condensation nuclei at 0.2% supersaturation (CCN, median value 180 $cm^{-3}$), specific humidity (q, median value 5 g $kg^{-1}$), cloud optical thickness (COT, median value 11). Note also an increase in precipitation frequency from 7% to 12% ($P_{freq}$). Results are obtained from 11-years data set featuring growing seasons, SMEAR II/MODIS.**

Page 13, Fig 4. It is interesting that there is no observed long-term trend in monoterpene concentrations during 2010-2022. Whilst this is a short period and the data is discontinuous I wonder if this can help say something about the sensitivity of monoterpene emissions to environmental drivers. Would theory/models expect an increasing trend over this short period? A short discussion on this might be useful.

**Thank you for the comment. We have published several papers where the relationships between environmental drivers and BVOC (incl. monoterpenes) emissions are elaborated either using the SMEAR II observations or indirectly with models and proxies obtained from the related parameters like oxidation products of monoterpenes. Discussion on the trends and drivers will be added to the revised paper.**

---

## Author Response (AR2)

Dear Editor,
Thank you for your final comments for the manuscript. We have made all the suggested corrections in this submitted manuscript.

Dear Authors,

There are some typos in the manuscript. I am highlighting below. Please make the corrections before uploading the manuscript for publication.

1. P3, L73, "in under development". "in" should be "is"
2. P3, L87, "unpected" should be "unexpected"
3. P4, L117. "participating to" be "participating in"
4. P12, L324. define "SA"
5. P12. L341. here are two "have". please remove one
6. P15, L421. "Eerth" to "Earth"
7. P21, L564. "whcih" to "which"

Best regards,
Xiaohong Liu